# Variant Characterization of a Representative Large Pedigree Suggests “Variant Risk Clusters” Convey Varying Predisposition of Risk to Lynch Syndrome

**DOI:** 10.3390/cancers15164074

**Published:** 2023-08-12

**Authors:** Mouadh Barbirou, Amanda A. Miller, Amel Mezlini, Balkiss Bouhaouala-Zahar, Peter J. Tonellato

**Affiliations:** 1Circulating Tumor Cell Core Laboratory, Population Science Division, Medical Oncology Department, Medical College, Sidney Kimmel Cancer Center, Thomas Jefferson University, Philadelphia, PA 19107, USA; amamiller@mcw.edu; 2Center for Biomedical Informatics, Department of Health Management and Informatics, School of Medicine, University of Missouri, Columbia, MI 65211, USA; tonellatop@health.missouri.edu; 3Medical School, University of Tunis El Manar, Tunis 1068, Tunisia; balkiss.bouhaouala@fmt.utm.tn; 4Medical Oncology Division, Salah Azeiz Oncology Institute, University of Tunis El Manar, Tunis 1068, Tunisia; amel.mezline@rns.tn; 5Laboratory of Venoms and Therapeutic Biomolecules, LR16IPT08 Institute Pasteur of Tunis, University of Tunis El Manar, Tunis 1068, Tunisia

**Keywords:** Lynch syndrome, colorectal cancer, “variant risk cluster” predisposition, familial germline variants, whole-genome sequencing

## Abstract

**Simple Summary:**

Approximately 20% of colorectal cancer (CRC) cases are diagnosed in individuals under 40, with a severe prognosis due to germline variant accumulation. Many of these variants have been classified as hereditary cancer causative, while others remain poorly researched. The identification of germline variants across different populations is critical for accurate prognosis, treatment, and follow-up. We aimed to identify and predict the functional implications of germline variants using whole-genome sequencing of a Tunisian pedigree with Lynch syndrome CRC risk. Two SNPs/indels were identified in genes with CRC association (MLH1 and PRH1-TAS2R14) and four in genes with non-CRC cancer association (PPP1R13B, LAMA5, FTO, and NLRP14). Three structural variants overlapped genes associated with non-CRC digestive cancer (RELN, IRS2, and FOXP1) and one overlapped RRAS2 with non-CRC cancer associations. This study provides further evidence of genetic predisposition according to the risk clustering of variants based on their functional implications and risk mechanisms within the same pedigree.

**Abstract:**

Recently, worldwide incidences of young adult aggressive colorectal cancer (CRC) have rapidly increased. Of these incidences diagnosed as familial Lynch syndrome (LS) CRC, outcomes are extremely poor. In this study, we seek novel familial germline variants from a large pedigree Tunisian family with 12 LS-affected individuals to identify putative germline variants associated with varying risk of LS. Whole-genome sequencing analysis was performed to identify known and novel germline variants shared between affected and non-affected pedigree members. SNPs, indels, and structural variants (SVs) were computationally identified, and their oncological influence was predicted using the Genetic Association of Complex Diseases and Disorders, OncoKB, and My Cancer Genome databases. Of 94 germline familial variants identified with predicted functional impact, 37 SNPs/indels were detected in 28 genes, 2 of which (MLH1 and PRH1-TAS2R14) have known association with CRC and 4 others (PPP1R13B, LAMA5, FTO, and NLRP14) have known association with non-CRC cancers. In addition, 48 of 57 identified SVs overlap with 43 genes. Three of these genes (RELN, IRS2, and FOXP1) have a known association with non-CRC digestive cancers and one (RRAS2) has a known association with non-CRC cancer. Our study identified 83 novel, predicted functionally impactful germline variants grouped in three “variant risk clusters” shared in three familiarly associated LS groups (high, intermediate and low risk). This variant characterization study demonstrates that large pedigree investigations provide important evidence supporting the hypothesis that different “variant risk clusters” can convey different mechanisms of risk and oncogenesis of LS-CRC even within the same pedigree.

## 1. Introduction

Colorectal cancer (CRC) is the most frequent neoplasm worldwide, accounting for 8% of cancer-related deaths [1,2]. Pathogenetic variants in known high-penetrance cancer-risk-associated genes have been implicated in up to 8% of all CRC cases, where one in five (20%) cases of this type were diagnosed in under 40 year olds [3,4,5] compared to the average worldwide age at diagnosis of 65 [6]. Familial association (defined to be families with at least two affected members) appears in an estimated 35% of all CRC cases [7]. The most common familial CRC is Lynch syndrome (LS) [8,9]. The familial forms of CRC genetic predisposition have been correlated with germline mutations or epimutations in mismatch repair (MMR) genes such as *MLH1*, *MSH2*, *MSH6*, and *PMS2* for nonpolyposis cases and in *APC* and *MUTYH* for Adenomatous colonic polyposis with recessive inheritance [4,5]. Despite the current knowledge of genetic predisposition in these hereditary forms of CRC (such as LS, Gardner syndrome, Juvenile polyposis coli, and others), much of the familial relationships and mechanisms of risk remain unexplained [10].

Genome-wide association via high throughput sequencing (HTS) technologies and the wide collection of variant functional predictive analytics, clouded applications, and databases are critical to the success of identifying new germline likely causative variants implicated in cancer predisposition. These powerful emerging tools help detect deleterious genomic changes in part responsible for the hereditary CRC development, diagnosis, predicted optimal treatment, and therefore long-term prognosis. Approximately 40% of patients with an inherited tumor syndrome exhibit a variant of uncertain significance, as revealed through sequencing analyses that examine germline variants involved in the production of truncated proteins and associated with alterations caused by hereditary pathologies at the germinal level [11]. These variants typically involve a single amino acid substitution, which cannot a priori be definitively classified as pathogenic or benign [12]. Conversely, synonymous nucleotide substitutions, which generally do not cause alterations in protein structure, have been found to be pathogenic in some instances, depending on their genomic location [13]. Additionally, variants appearing together in the same gene or different genes may coexist and co-segregated with the disease phenotype within a single family, potentially explaining the correlated predisposition risk of the family. In some cases, these variants contribute more significantly to cancer risk than classic pathogenic Mendelian variants, and when implicated in tumor predisposition, can cooperatively contribute to an increased risk of cancer development as low-risk alleles [14,15]. However, HTS studies have not covered all such cancers and therefore, it is highly likely that functionally pertinent variants and mutations and genes conveying predisposition to CRC and LS are yet to be discovered. This gap in the current knowledge of familial forms of CRC such as LS requires further clinical evaluation of hereditary CRCs supported by germline studies of familial cases [16].

In this study, we aim to identify, annotate, and computationally predict the functional implication of previously known and novel germline SNPs, indels, and structural variants using a whole-genome sequencing approach of a Tunisian large pedigree with three familiarly grouped members affected or at-risk to LS-CRC.

## 2. Materials and Methods

### 2.1. Sample/Data Collection

An LS-affected, large-pedigree Tunisian family with 37 total and 12 known affected members was recruited for this study (Figure 1). Peripheral blood from 11 members (oval circled) was collected. Individual subjects’ clinical, environmental, and behavioral data were collected from medical records and personal interviews based on an interrogatory form conducted by the study personnel.

Lynch syndrome status within this family was tested and confirmed during routine clinical work. The three patients collected from in the HRLS group were classified as meeting both the Amsterdam criteria II that have been established by the International Collaborative Group on HNPCC for assistance in identifying Lynch syndrome [17,18] and the original Bethesda guidelines [19]. Subsequent blood germline testing using PCR amplification and direct sequencing of the entire coding region and the exon–intron boundaries for MMR genes (MLH1, MSH2, and MSH6) [20,21] revealed a single deleterious germline alteration affecting the *MLH1* gene (mutation c.-168_c.116 + 713 del). This corresponds to a 997 bp deletion that encompasses the entirety of exon 1, a portion of intron 1, and a section of the *MLH1* promoter, and was observed in all subjects within the HRLS group who underwent germline testing.

Data summarized in Table 1 were recorded in a study database and maintained in a secure, private manner consistent with the Declaration of Helsinki and the permission of Salah Azaiz Institute Ethics Committee registration number: ISA/2016/02. All subjects were informed about the purposes of the study and consented in writing to participate in the study. The 11 subjects were stratified into three groups based on familial cancer status. The high risk to LS group (HRLS, Figure 1, red ovals) are affected subjects of the pedigree; intermediate risk to LS (IRLS, Figure 1, green ovals) includes CRC-free subjects that have at least one affected parent, and low risk to LS (LRLS, Figure 1, blue ovals) are those with no relatives affected in the subject’s immediate triplet (subject and both parents).

### 2.2. DNA Extraction and Quality Assessment

Genomic DNA was extracted from the 11 blood samples according to the manufacturer’s recommendation using a Flexigene DNA Whole Blood Kit (Qiagen, Hilden, Germany). DNA quality and quantity were assessed using a Qubit fluorometer (Invitrogen, Carlsbad, CA, USA) and electrophoresis migration in agarose gel 1%. The genomic DNA with good quality was subjected to library preparation prior to sequencing.

### 2.3. Whole Genome Sequencing (WGS)

Libraries were prepared using Nextera XT kit (Illumina, San Diego, CA) and pair-end sequencing (2 × 300 base pairs) with the Miseq Reagent V3 kit (Illumina) following the manufacturer’s instructions. The Nextera enzyme mix was used to simultaneously fragment input DNA and tag with universal adapters in a single tube reaction. Library purification was performed by Agincourt AMPure XP beads (Beckman Coulter, IN, USA) and Bioanalyzer (Agilent, Wilmington, DE, USA) was used for quantification and quality checking [22]. Libraries were sequenced using the Illumina NextSeq500 platform (Illumina Inc., San Diego, CA, USA). A total of 1564.4 GB with 142.22 GB on average per sample of raw data was generated on the sequencer, resulting in a mean sequence coverage depth of 45.88-fold (range of 37.71- to 59.17-fold).

### 2.4. Bioinformatic Variant Analysis (BVA)

The full bioinformatics variant analysis (BVA) is described in Figure 2 and the Appendix A. We used a novel functional implicated variant pipeline created in our previous work on breast cancer [23], modified to account for the pedigree relationship between subjects and familial and thus the risk-related nature of the detected variants. Briefly, for each subject’s high-throughput sequencing (HTS) sequence, alignment, poor-quality read filtering, single nucleotide polymorphisms (SNPs), insertions/deletions (indels), and structural variants (SVs) were called and variant quality score recalibration and filtering, annotation and removal of common and likely non-functional variants, and assessment of cancer-associated genes were performed.

## 3. Results

### 3.1. Clinical Characterization of the Pedigree Members

The large Tunisian pedigree (Figure 1) includes 39 members across four generations; 12 of them were diagnosed with LS CRC. In this study, 11 members’ (3 HRLS, 4 IRLS, and 6 LRLS) germline genomes were fully sequenced and their medical, environmental, and behavioral data were carefully analyzed. Individual and familial characterizations are described in Table 1. The HRLSs presented an average age of 48 ± 12.16 and BMI of 26.08 ± 3.11, IRLSs had an average age and BMI of 38.25 ± 17.63 and 26.77 ± 3.47, respectively, and the LRLSs had an average age and BMI of 35.50 ± 24.11 and 27.52 ± 2.64, respectively. Male sex distribution was 33.34% (1/3) HRLS, 0% (0/4) IRLS, and 50% (3/6) LRLS. Concerning lifestyle, 66.66% of HRLS recorded high brine and meat consumption, 100% of IRLS had a high fat consumption, and 75% of LRLS presented high meat and fat consumption. A total of 66.66% of HRLS were tobacco and alcohol users, 25% of IRLS were tobacco users, and 50% of LRLS were tobacco and alcohol users. A total of 66% of HRLS had hyperglycemia and hypertension, 75% of IRLS had hypertension and 50% had hyperglycemia, and 25% of LRLS had hypertension. An analysis of the clinical characteristics of each group found no statistically significant associations.

### 3.2. Variant Characterization of the Pedigree

#### 3.2.1. SNPs and Indels

Before filtering, 7,836,438 SNPs and 2,039,903 indels were detected across the 11 genomes. A total of 23.63% (2,334,312 of 9,876,341) of all variants satisfied the low-frequency threshold (<0.01 AF 1000 G ALL and non-TCGA ExAC ALL) and 3.46% (80,905 of 2,334,312) of the variants demonstrated a probable deleterious function (CADD scaled score > 10). Subsequent annotation stratified the likely functionally implicated variants into coding (2824) and non-coding (78,081) variants. Lastly, filtering for variants predicted deleterious by having at least three of MutationTaster, PolyPhen V2, Provean, and SIFT resulted in 1961 coding and 9826 non-coding predicted deleterious variants. From the 1961 low-frequency, predicted functionally deleterious coding variants, 79 were found in all 11 sequenced individuals: 37 in all HRLS members, 27 in the IRLS group, and 15 in the LRLS. Of the 9826 non-coding filtered variants, all 9826 were found in members of at least one group; 4171 of the 9826 were found in each HRLS member, 2820 were found in every IRLS member, and 2835 in every LRLS member. In addition, 19 non-coding variants were exclusively detected only in a particular group of the pedigree: 2 variants exclusive to members in HRLS, 3 exclusives to IRLS, and 14 exclusives to LRLS, and 4 of these had a RegulomeDB Score < 4. Most of the detected non-coding variants were Intergenic (37,412) and found in all samples, whereas 30344 Intronic variants were found in all samples, and only 7 were found in samples in the same groups (Table 2).

#### 3.2.2. SNP and Indels with Evidence of Familial and Risk-Implicated Genes

Of the 83 variants identified in the cohort, 39 genes were identified containing at least one variant. *PABPC3* had four variants (rs79397892, rs78826513, rs78552667, and rs80261016) found in all samples and one variant (rs201411821) found in all high and intermediate risk samples (HRLS and IRLS). Three of these five variants (rs78826513, rs201411821, and rs80261016) were classified as oncogenic driver variants according to SNPnexus (“Driver” as defined in the oncogenic classification by Cancer Genome Interpreter). *KRT18* had four variants found in all samples, though none satisfied the oncogenic (or “Driver” predicted as tumor driver according to Cancer Genome Interpreter) threshold. One additional variant (rs201602708) located in *MACF1* did not satisfy the oncogenic threshold. Five of the detected variants were in different genes that were previously described as common mutations in a collection of cancers. rs63750539 in the *MLH1* gene has been described in several types of cancer including CRC and LS, rs373141354 in the *PPP1R13B* gene is associated with melanoma according to the Genetic Association of Complex Diseases database, rs551763507 in the *LAMA5* gene is correlated to neuroblastoma, rs76670455 in *NLRP14* gene to leukemia, and rs763119571 in *TAS2R19* is also described in CRC according to the Cancer Genome Interpreter database. Concerning non-coding variants, four variants belonging to different genes (rs544153916 in *PODN*, rs116197074 in *SCP2*, and rs116526711 in *MAML3*) were noted only in samples from the LRLS group, and one of these variants rs115378978 in the *FTO* gene was previously associated with prostate cancer according to Disorders and SNPnexus databases (Table 3).

#### 3.2.3. Structural Variants (SVs)

A total of 11,171 SVs were found via smoove. Of the total SVs detected, their classifications were 4120 deletions, 194 duplications, 6560 breakends, and 297 inversions. Filtering for low-frequency novel variants resulted in 1122 deletions, 105 duplications, 5492 breakends, and 137 inversions. Filtering by highly likely functional (AnnotSV ranking > 3) resulted in 164 deletions, 17 duplications, 274 breakends, and 18 inversions. Finally, filtering for total length (>= 50 bp) resulted in 140 (of the 164) deletions. A familial analysis was performed to test sharing across study groups found, finding 154 breakends, 133 deletions, 4 duplications, and 13 inversions in each of the 11 subjects. A total of 35 shared breakends (23 shared within HRLS, 4 in IRLS, and 8 in LRLS) and 18 shared deletions (7 in HRLS, 4 in IRLS, and 7 in LRLS). Two duplications were shared by all members of LRLS and one inversion in IRLS and one inversion in LRLS. AnnotSV analysis for the SVs genomic location showed that 26 breakends and 17 deletions were intronic variants, but the 2 noted inversions were Transcript Start-Transcript End variants and the only detected duplication was an intronic variant (Table 4).

#### 3.2.4. SVs with Risk-Implicated Genes

The 48 identified SVs were evaluated for functional prediction; the results showed that 43 SVs overlapped with genes with a probable functional impact, and 9 SVs overlapped with genes with an unlikely functional impact. Of these 43 SVs of probable impactful, 5 SVs overlapped with 4 genes with likely impact, 4 in the HRLS group (2 SVs in *RELN*, 1 each for *FOXP1* and *RRAS2* genes), and 1 in the IRLS group for the *IRS2* gene previously associated with multiple cancers, including CRC according to OncoKB Cancer Genes list database. Two of the forty-three breakend SVs found in all members of the LRLS group had a potential impact on *RELN*, a gene correlated to several cancers including gastric cancer according to OncoKB Cancer Genes list database. One of the forty-three duplication SVs in all members of the IRLS group contains *IRS2*, a gene associated with esophageal, intestinal, stomach, and CRC cancers according to the My Cancer Genome database. One deletion SV in all members of LRLS impacts *FOXP1*, a gene correlated to several cancers including esophagogastric, gastrointestinal, and CRC in the My Cancer Genome database. Finally, one breakend SV found in all members of HRLS affects *RRAS2*, a gene associated with both breast and ovarian cancers according to the My Cancer Genome database (Table 5).

## 4. Discussion

Several studies have examined the complex molecular heterogeneity of LS CRC in large families. The literature suggests that LS is caused by genetic and epigenetic variants sporadically found in genes, such as MMR (*MLH1*, *MSH2*, *MSH6*, *PMS2,* and *EPCAM*) associated with flat intra-mucosal neoplastic lesions [24,25]. However, a recent study by Binder et al. defined a third pathway for LS and showed the existence of two distinct genetic subtypes of the LS CRC [26]. These recent findings suggest that the risk and genesis of LS CRC may be caused by various multiply expressed functionally important “variant risk clusters” of germline mutations, each cluster independently associated with various pathways to carcinogenesis, and in a similar manner, each cluster may define both the type and degree of risk to LS CRC even within one large pedigree. The evidence of our results coupled with the broader collection of results in the literature supports such a working hypothesis.

These observations lead to the hypothesis that, in addition to the relatively well-characterized *MMR* deficiency in LS, other germline mutations or groups of mutations may contribute to the disruption of previously unassociated pathways, thus being associated with varying risks and oncogenesis of LS CRC. In this study, we investigated the germline mutational profiles of a large pedigree Tunisian family with Lynch syndrome-associated colorectal cancer using high throughput whole-genome sequencing. Subjects were grouped by familial cancer status into high, intermediate, and low risk to LS. Overall, we identified 94 germline variants, including 11 novel and rare cancer pathogenic variants previously described in cancer. Then, we clustered the identified germline variants according to the pedigree risk status into LRLS, IRLS, and HRLS groups.

The high-risk LS patients shared a missense mutation (rs63750539, p.Ala111Val) in *MLH1* which is unlikely to be the cause of LS-CRC predisposition.

The *MLH1* gene has been established as a causative gene for LS and presents the highest risk of CRC among individuals over 75 (46.6% of women and 51.4% of men) who are affected by the *MLH1* variation. Rates range from 0% (at age 30) to 48.3% (at age 75) in females, and from 4.5% (at age 30) to 57.1% (at age 75) in males [22,23]. In another study, *MLH1* variants were correlated with the highest risk of developing CRC in both heredity and sporadic cases [27]. Furthermore, two *MLH1* 5′UTR variants (c.-28A > G and c.-7C > T) were associated with early-onset CRC [28]. In addition, a cohort study reported that *MLH1* is the most frequently mutated gene in early-onset sporadic CRC patients, exhibiting four pathogenic variations: c.C793T (p.R265C), c.C1029A (p.Y343X), c.C793T (p.R265C), and c.C1029A (p.Y343X) [29]. On the other hand, recent studies have found that promoter methylation of the *MLH1* gene is prone to be silenced in CRC carcinogenesis pathways, and around 50% of *MLH1*-deficient tumors exhibit *MLH1* promoter methylation [30]. Moreover, *MLH1* promoter methylation in CRC cases was highly correlated with a BRAF V600E somatic mutation [31]. This variant was clinically classified as a hereditary sequence variant identified in disease-related genes directly affecting the clinical management of patients with LS-CRC [32]. To better understand the high risk effect on the pedigree, we investigated known driver mutations that are likely related to *MMR* deficiency. First, we investigated mutations in genes that play a key role in the adenoma–carcinoma model of CRC such as *APC*, *KRAS*, *TP53,* and binding/transactivated genes. We found the rs373141354 variant in the *PPP1R13B* gene (p.Gly866Arg), which assists *TP53* activation during the cell apoptosis and lowers their ability shared by all subjects HRLS group [33]. Although *TP53* is known to be rarely mutated in LS [34], our findings can be attributed to the high efficiency of *TP53* in maintaining genomic integrity by arresting cells with mutated or damaged DNA in the G1 phase of the cell cycle to enable the repair mechanism or induce the apoptosis pathway [35]. The balance between cell cycle arrest and induced apoptosis depends on *TP53* efficiency, which is related to the *PPP1R13B* activity identified in our research. Hence, we suggest that the variant rs373141354 (p.Gly866Arg) is associated with an increased risk of LS-CRC due to its low efficiency in cell cycle arrest.

Concerning the detected SVs overlapped with genes that have been previously correlated with different types of cancers, including digestive cancers, four shared SVs were found among HRLS subjects. Two SVs (7_103463079_103463080_BND_1 and 7_103463462_103463463_BND_1) were found to be related to the intronic region of the *RELN* gene, which has been correlated to several cancer risks including gastric cancer [36]. The large CpG islands are located at *RELN* promoter sites, and their transcriptional silence has been shown to be strongly controlled by promoter hypermethylation [37]. Consequently, a relationship between SVs and DNA methylation in cancer is speculated. Recent studies suggest that somatic copy number alterations in cancer are associated with DNA methylation [38], and numerous studies demonstrate that SVs may have a causal role in regulating CpG methylation [39,40]. Conversely, it is also possible that methylation could lead to SV imbalance by increasing DNA breaking [41,42]. Our observation enhances our growing understanding of the relationship between genetic (SVs) and epigenetic variation in cellular phenotype and the mechanism of gene regulation as well as the traits underlying the evolution of cancer with the presence of SVs in specific genome sites. The 71242366_71242638_DEL_1 deletion was noted in the *FOXP1* gene. A large amount of substantial evidence has demonstrated that the tumor microenvironment is closely linked to the initiation, promotion, and progression of CRC through various mechanisms, such as immune suppression and the angiogenesis process [43]. Variation in *FOXP3*, an intracellular key molecule for Treg development and function, has been associated with a dysregulation in subverting antitumor immune responses and promoting tumor progression. The last SV detected, 11_14348706_14348707_BND_1, was related to the intronic region of the *RRAS2* gene, which has previously been described in breast and ovarian cancers [44]. The role of LS in ovarian cancer was established and widely accepted, but the long-standing question of whether breast cancer should also be included under the umbrella of LS is still debated [45,46]. A recent study, consistent with other studies, shows that carriers of LS mutations tend to have earlier manifestations of breast cancer [47].

Regarding the IRLS group, a unique missense variant rs551763507 (p.Gly3688Glu) in the *LAMA5* Laminin gene was identified in all subjects of the group. Based on Laminin’s function, these variants are not the most probable candidates to play a role in CRC susceptibility [48]. However, recent studies have identified *LAMA5* in orthotropic metastases. The expression of Laminin 511 was associated with the upregulation of a set of genes regulating angiogenesis in TCGA data [48,49]. The Gordon group has demonstrated that the Laminin chains are localized within the vascular basement membrane on the basolateral surface of cancer cells, while colonic epithelial cells normally do not express vascular basement membrane Laminins. Consequently, the profound effect on vascular morphology and function upon *LAMA5* mutation and inhibition of *LAMA5* expression specifically by colon cancer cells indicates that cancer cell Laminin 511 deposition is important for colonic cells, promoting angiogenesis. Our results may suggest that inhibiting the production of vascular basement Laminins by tumor cells may serve as an efficient approach to prevent growth and the ability of tumor cells to regulate angiogenesis. Another novel variant was identified as an IRLS member as rs76670455 in the *NLRP14* gene. *NLRP14* has been described as a negative regulator of IFN responses. Interestingly, as inflammatory signaling pathways contribute to B cell lymphoma transformation, it is tempting to speculate that *NLRP14* might contribute to cancer [50]. Another interesting aspect of NLR proteins is their expression in a panel of immune cells, notably myeloid cells and B cells, and their function as a negative regulator of inflammatory responses [51]. This differential expression of *NLRP14* might be involved in the malignant transformation process. For the noted SVs in this HRLS pedigree group, only one variant was detected, the 13_110418067_110419056_DEL_1 in the intronic region of the *IRS2* gene, which was shared by all IRLS subjects and has been described as related to several digestive cancers. Over-expression of *IRS2* increases CRC cell adhesion to a similar extent as IGF-1 stimulation. Changes in adhesion, both increasing and decreasing, are important properties of metastasizing cancer cells and are involved in the invasion process, migration, and distant seeding of tumors [52]. In addition, it has been proven that the *PI3K* pathway is frequently dysregulated during CRC progression [53]. The TCGA Network demonstrated that high levels of *IRS2* expression are mutually exclusive with *IGF2* over-expression and with other mutations in the PI3K pathways in CRC. This suggests that the over-expression of *IRS2* may be one mechanism by which the *PI3K* pathway could be dysregulated in CRC [54]. In summary, *IRS2* appears to be a potential candidate as an oncogene driver, and the *IGF1R*-*IRS2*-*PI3K* axis could be an important therapeutic target in CRC.

In connection with variants specific to LRLS, we noted rs763119571 in the *TAS2R19* gene with a deleterious function. Previous research has shown the association of this variant with CRC risk [24]. Genetic variants in type 2 bitter taste receptors (TAS2R) may influence health-related outcomes and are expressed within the oral cavity [55,56], the gastrointestinal mucosa [57], and the lungs [58]. TAS2R variants are hypothesized to play roles in an individual’s food preferences [59] and the neutralization and expulsion of toxins from the colon/rectum [60], thereby influencing cancer risk. The last noted variant with the deleterious function was rs115378978 in the *FTO* gene. Different polymorphisms of the *FTO* gene have been consistently associated with obesity. However, recent genome studies reveal that genetic variants in this gene are associated not only with human adiposity and metabolic disorders but also with several cancers, including colorectal cancer, since they can activate several signaling and hormonal pathways to increase cancer incidence. The hormones included in this carcinogenesis process could be ghrelin, oxytocin, and Leptin. Hence, *FTO* polymorphisms could exert an influence on the hormonal balance and physiologic factors and might increase cancer risk [60]. The variants identified in *FTO*, *TAS2R*, and *NLRP14* genes were correlated with lifestyle-related factors for cancer installation, which are expected to be found in cases belonging to the LRLS group who are CRC-free. No oncogenic impactful SVs were detected in the LRLS group, which may explain the low risk of these subjects developing LS CRC and the crucial role of these variants in heredity cancer development.

Our current findings clearly illustrate that subtle, familiarly grouped genetic factors underlying risk to LS CRC extend beyond the well-documented familial CRC syndrome genes. Our data suggest that comprehensive germline testing in all LS CRC patients will provide comprehensive results to identify substantially more opportunities for robust and accurate genetically driven cancer characterization and subsequent prevention than established in current practice. Moreover, the magnitude of developing secondary cancer (breast or ovarian) is still unknown. Hence, our study sheds light on the importance of risked grouped germline screening to identify secondary risks and even predict the site of metastases during CRC progression.

This study’s limitations include the lack of testing on all family members. On the other hand, given the limited studies on SV identification, the impact of the SVs found in our study was predicted based on the hypothesis and recent functional studies. Thus, further studies in many LS-CRC families are needed to confirm the effect of the 57 SVs identified in our study. In addition, despite the extensive advantages of whole-genome sequencing and data processing, there remain several gaps in the technology and analysis. For example, the restricted resolution is caused by limitations in the read depth related to the quality of aligned sequences; a higher depth provides a greater power to call and identify new variants [61]. However, our mixed pipeline generated for our WGS-specific study allows us to detect new and known coding and non-coding variants across the whole genome of the 11 sequenced individuals, greatly facilitating the germline profile evolution of LS-CRC pedigrees and opening new opportunities for cancer pedigree studies. The introduction of several commercially available multigene panels has tremendous promise for clinical use but simultaneously raises weaknesses such as clear criteria for selecting above average risk patients to undergo such clinical panel testing [62] and the optimal choice of the prescribed panel [63]. Several multigene panels are available for hereditary CRC and the National Comprehensive Cancer Network (NCCN) has provided a useful protocol for predicting the appropriate panel according to the family history of patients to better optimize the patient care [64]. A limitation of our study is that not all members of the family were tested for the *MLH1* gene variant (c.-168_c.116 + 713del) detected in the family. This may lead to the study being considered a population-based WGS investigation in the context of *MLH1* carriers and non-carriers, potentially weakening our conclusions about genetic modifiers in LS.

## 5. Conclusions

In this study, we analyzed the germline landscape of a Tunisian family with a predisposition to LS CRC and identified a total of 94 germline variants affecting 39 genes, 6 of which have been previously described in cancer, and 57 SVs, 48 of which were related to 43 genes, with 4 of these genes categorized as oncogenes. According to the familial definition of LS risk in the pedigree members, we identified three “variant risk clusters” associated with the high, intermediate, and low LS CRC risk groups in the pedigree. The results showed that variants related to high-risk LS members may be causative of the disease, while other variants present in intermediate-risk members may develop LS through very different mechanistic disruptions and low-risk members with these variants may not develop LS at all. The application of HTS technology with such variant clustering for germline screening will efficiently provide further insights into the etiology of hereditary cancer and a huge opportunity to improve LS clinical suspicion. The significance of germline variants in cancer predisposition is still poorly explored, and this study contributes to filling this knowledge gap.

## Figures and Tables

**Figure 1 cancers-15-04074-f001:**
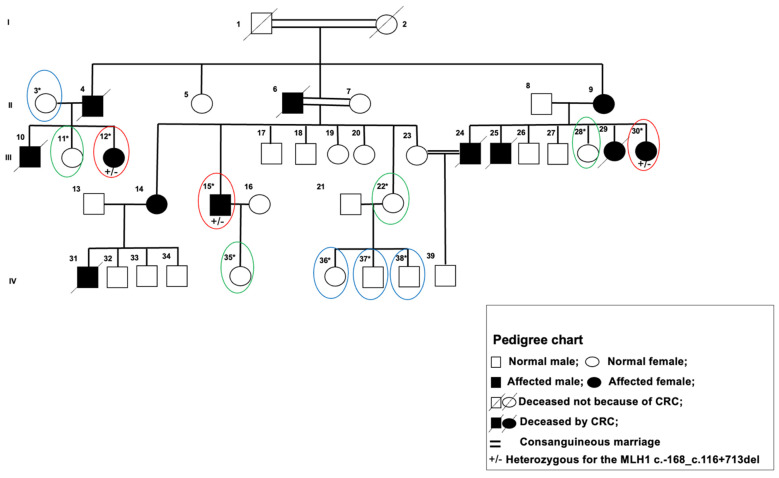
A pedigree of predisposition to colorectal cancer Lynch syndrome. I, II, III, IV the number of generations. * Sequenced subjects; Group 1 (high risk to LS group): subject and parent affected by LS-CRC, red circles; Group 2 (intermediate risk to LS): subject LS-CRC free and one of the parents is affected, green circles; Group 3 (low risk to LS): subject and both parents are LS-CRC free, blue circles.

**Figure 2 cancers-15-04074-f002:**
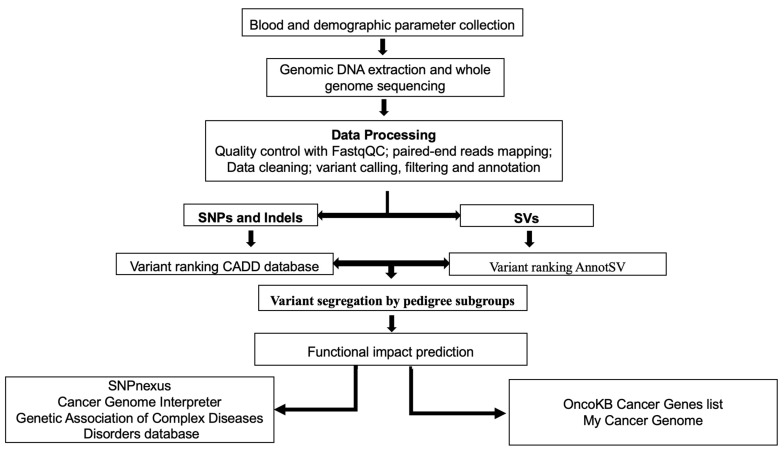
Schematic representation of the germline variant prioritization workflow. WGS: whole genome sequencing, QC: quality control, SNPs: single nucleotide polymorphisms, indels: insertions/deletions, SVs: structural variants, CADD: combined annotation dependent depletion, OncoKB: Oncology Knowledge Base.

**Table 1 cancers-15-04074-t001:** Demographic and clinical characteristics of the included family members.

Parameters	HRLS N = 3 (%)	IRLS N = 4 (%)	LRLS N = 4 (%)
Gender (Males/Females)	1/2	0/4	2/2
Age (years) ^1^	48 ± 12.16	38.25 ± 17.63	35.50 ± 24.11
BMI ^1^	26.08 ± 3.11	26.77 ± 3.47	27.52 ± 2.64
Vegetable consumption			
High	0 (0)	3 (75)	4 (100)
Low	3 (100)	1(25)	0 (0)
Brine consumption			
High	2 (66.66)	0 (0)	1 (25)
Low	1 (33.34)	4 (100)	3 (75)
Meat consumption			
High	2 (66.66)	2 (50)	3 (75)
Low	1 (33.34)	2 (50)	1 (25)
Fat consumption			
High	0 (0)	4 (100)	3 (75)
Low	3 (100)	0 (0)	1 (25)
Smoking			
Never used	1 (33.34)	3 (75)	2 (50)
Tobacco users	2 (66.66)	1 (25)	2 (50)
Alcohol			
Never drink	1 (33.34)	4 (100)	2 (50)
Alcohol users	2 (66.66)	0 (0)	2 (50)
Physical activity level			
High	0 (0)	2 (50)	4 (100)
Low	3 (100)	2 (50)	0 (0)
Medical History			
Hypertension	2 (66.66)	3(75)	1(25)
Hyperglycemia	2 (66.66)	2(50)	0 (0)
Anemia	0 (0)	0 (0)	0 (0)

BMI: Body mass index; tobacco users: former users and current users; alcohol users: former drinkers and current drinkers, ^1^: mean ± standard deviation; high risk to LS group (HRLS); intermediate risk to LS (IRLS); low risk to LS (LRLS).

**Table 2 cancers-15-04074-t002:** Variants count for the whole pedigree subjects and shared variants by subjects under the same pedigree group with the functional annotation of noncoding variants according to ANNOVAR.

Variants Filtering	Variant Count	
VQSR	9,876,341 (7,836,438 SNPs and 2,039,903 Indels)	
<0.01 AF 1000 G ALL and non-TCGA ExAC ALL	2,334,312	
CADD (SNPs) or CADD Indel (Indels) Scaled Phred Score > 10	80,905	
Variant stratification	Coding Variants	Non-Coding Variants
Total count	2824	78,081
Predicted deleterious by having at least three of MutationTaster, PolyPhen V2, Provean, and SIFT	1961	9826
Shared by all samples in a group	79 (HRLS: 37; IRLS: 27; LRLS: 15)	9826 (HRLS: 4171; IRLS: 2820; LRLS: 2835)
Exclusive to a particular group	0	19 (HRLS: 2; IRLS: 3; LRLS: 14)
RegulomeDB Score < 4	NA	LRLS: 4
Functional annotation of noncoding variants according to ANNOVAR		
Variants annotation according to region hit from RefSeq	Variants shared by all subjects	Variants shared by subjects in the same group
Intergenic	37,412	10
Intronic	30,344	7
ncRNA_intronic	5072	0
3′UTR	2169	0
Upstream and Downstream	1825	2
5′UTR5	768	0
ncRNA_exonic	447	0

RefSeq: reference sequence database; ncRNA: non-coding transcript variant; NA: not applicable; VQSR: Variant Quality Score Recalibration; ExAC: exome aggregation consortium; AF: allele frequency; 1000 G: 1000 genomes project for all individual in this release; CADD: combined annotation dependent depletion; SNPs: single nucleotide polymorphisms; indels: insertions/deletions; PolyPhen V2: PolyPhen Version 2; high risk to LS group (HRLS); intermediate risk to LS (IRLS) low risk to LS (LRLS); TCGA: The Cancer Genome Atlas Program; SIFT: sorting intolerant from tolerant; PROVEAN: Protein Variation Effect Analyzer.

**Table 3 cancers-15-04074-t003:** Classification of detected variants by gene and cancer impact according to pedigree groups.

Coding Variants										
Genes	Detected Variants Stratified by Pedigree Groups				AF	Functional Annotation	Cancer Related	CRC Related	Oncogenic Classification	Tumor Driver
	SNP ID	HRLS	IRLS	LRLS						
*PABPC3*	Variants Count	5	5	4						
	*rs79397892*	Yes	Yes	Yes	0.0054	EX	-	-	Passenger	True
	*rs78826513*	Yes	Yes	Yes	NA	EX	-	-	Driver	True
	*rs78552667*	Yes	Yes	Yes	NA	EX	-	-	Passenger	True
	*rs201411821*	Yes	Yes	-	NA	EX	-	-	Driver	True
	*rs80261016*	Yes	Yes	Yes	NA	EX	-	-	Driver	True
*KRT18*	Variants Count	4	4	4						
	*rs78343594*	Yes	Yes	Yes	NA	EX	-	-	Passenger	False
	*rs77999286*	Yes	Yes	Yes	NA	EX	-	-	Passenger	False
	*rs75441140*	Yes	Yes	Yes	NA	EX	-	-	Passenger	False
	*NA*	Yes	Yes	Yes	NA	EX	-	-	NA	NA
*CNN2*	Variants Count	2	2	1						
	*rs77830704*	Yes	Yes	Yes	NA	EX	-	-	Passenger	False
	*rs75676484*	Yes	Yes	-	NA	EX	-	-	Passenger	False
*SLC25A5*	Variants Count	1	2	1						
	*rs753913830*	Yes	Yes	Yes	NA	EX	-	-	Passenger	False
	*rs199707714*	-	Yes	-	NA	EX	-	-	Passenger	False
*MYH13*	Variants Count	1	1	1						
	*rs186137259*	Yes	Yes	Yes	﻿0.0016	EX	-	-	Passenger	False
*DNAH2*	Variants Count	1	1	1						
	*rs140035206*	Yes	Yes	Yes	﻿0.0022	EX	-	-	Passenger	False
*ANP32B*	Variants Count	1	1	-						
	*rs76167314*	Yes	Yes	-	NA	EX	-	-	Passenger	False
*CNKSR1*	Variants Count	1	1	-						
	*rs140685957*	Yes	Yes	-	NA	EX	-	-	Passenger	False
*CTNNBIP1*	Variants Count	1	1	-						
	*rs138271667*	Yes	Yes	-	0.0018	EX	-	-	Passenger	False
*DNAH3*	Variants Count	1	1	-						
	*rs147732992*	Yes	Yes	-	0.0034	EX	-	-	Passenger	False
*KDM1A*	Variants Count	1	1	-						
	*rs144822945*	Yes	Yes	-	0.0015	EX	-	-	Passenger	False
*PRSS3*	Variants Count	2	1	1						
	*rs141382822*	Yes	Yes	Yes	NA	EX	-	-	Passenger	False
	*rs751456445*	Yes	-	-	NA	EX	-	-	Passenger	False
*AATK*	Variants Count	1	-	-						
	*rs61738829*	Yes	-	-	0.0088	EX	-	-	Passenger	False
*ALPK3*	Variants Count	1	-	-						
	*NA*	Yes	-	-	NA	EX	-	-	NA	NA
*ANKRD34B*	Variants Count	1	-	-						
	*rs145614517*	Yes	-	-	0.0030	EX	-	-	Passenger	False
*ATXN2*	Variants Count	1	-	-						
	*rs374319477*	Yes	-	-	NA	EX	-	-	Passenger	False
*CUX2*	Variants Count	1	-	-						
	*rs202242120*	Yes	-	-	0.0040	EX	-	-	Passenger	False
*ERAP2*	Variants Count	1	-	-						
	*rs145045143*	Yes	-	-	0.0010	EX	-	-	Passenger	False
*FAM136A*	Variants Count	1	-	-						
	*rs80277652*	Yes	-	-	NA	EX	-	-	Passenger	False
*MACF1*	Variants Count	1	-	-						
	*rs201602708*	Yes	-	-	0.0002	EX	-	-	Driver	True
** *MLH1* **	Variants Count	1	-	-						
	** *rs63750539* **	**Yes**	**-**	**-**	**NA**	**EX**	**Leukemia; Lymphoma;** **Melanoma; Pancreatic;** **Liver; Fallopian; Endometrial; Ovarian; Breast; Prostate; Bladder; Thyroid; Esophageal; Lung**	**Lynch syndrome; Colorectal melanoma; Microsatellite Instability; Hereditary non-polyposis; Colon; Gastric;** **Stomach**	**Driver**	**True**
*PCSK5*	Variants Count	1	-	-						
	*rs372055352*	Yes	-	-	NA	EX	-	-	Passenger	False
*PLA2G6*	Variants Count	1	-	-						
	*NA*	Yes	-	-	NA	EX	-	-	NA	NA
** *PPP1R13B* **	Variants Count	1	-	-						
	** *rs373141354* **	**Yes**	**-**	**-**	**NA**	**EX**	**Melanoma**	**-**	**Passenger**	**False**
*RIMKLA*	Variants Count	1	-	-						
	*rs34142209*	Yes	-	-	0.0082	EX	-	-	Passenger	False
*RNF207*	Variants Count	1	-	-						
	*NA*	Yes	-	-	NA	EX	-	-	NA	NA
*SIPA1L3*	Variants Count	1	-	-						
	*rs201766021*	Yes	-	-	0.0002	EX	-	-	Passenger	False
*XIRP1*	Variants Count	1	-	-						
	*rs147417919*	Yes	-	-	0.0032	EX	-	-	Passenger	False
*CD8A*	Variants Count	-	1	-						
	*rs200750291*	-	Yes	-	0.0012	EX	-	-	Passenger	False
** *LAMA5* **	Variants Count	-	1	-						
	** *rs551763507* **	**-**	**Yes**	**-**	**NA**	**EX**	**Neuroblastoma**	**-**	**Passenger**	**False**
*MRPL24*	Variants Count	-	1	-						
	*rs561581574*	-	Yes	-	NA	EX	-	-	Passenger	False
** *NLRP14* **	Variants Count	-	1	-						
	** *rs76670455* **	**-**	**Yes**	**-**	**0.0058**	**EX**	**Leukemia**	**-**	**Passenger**	**False**
*TAAR5*	Variants Count	-	1	-						
	*rs9493386*	-	Yes	-	0.0026	EX	-	-	Passenger	False
** *PRH1-TAS2R14* **	Variants Count	-	-	1						
	** *rs763119571* **	**-**	**-**	**Yes**	**NA**	**EX**	**-**	**Colorectal**	**Passenger**	**False**
*TMEM8B*	Variants Count	-	-	1						
	*rs148540551*	-	-	Yes	0.0014	EX	-	-	Passenger	False
Noncoding variants										
*PODN*	Variants Count	-	-	1						
	*rs544153916*	-	-	Yes	0.0002	NA	-	-	Not protein-affecting	False
*SCP2*	Variants Count	-	-	1						
	*rs116197074*	-	-	Yes	0.0056	INT	-	-	Not protein-affecting	False
*MAML3*	Variants Count	-	-	1						
	*rs116526711*	-	-	Yes	0.007	INT	-	-	Not protein-affecting	False
** *FTO* **	Variants Count	-	-	1						
	** *rs115378978* **	**-**	**-**	**Yes**	**0.0078**	**INT**	**Prostate**	**-**	**Not protein-affecting**	**False**

AF: 1000 G Phase 3 all population allele frequency; row in bold: variant previously described as associated with cancer; CRC: colorectal cancer; SNP: single nucleotide polymorphism; ID: identification; rs: reference SNP; INT: intronic; EX: exonic; NA: not applicable; high risk to LS group (HRLS); intermediate Risk to LS (IRLS); low risk to LS (LRLS).

**Table 4 cancers-15-04074-t004:** Variants count for the whole pedigree subjects and shared variants by subjects under the same pedigree group with variants location according to AnnotSV.

Variants Filtering	Variant Count			
Deletions and duplications: duphold depth based	4314 (4120 DEL, 194 DUP)			
Breakends and inversions: NA	6857 (6560 BND, 297 INV)			
Total count	BND	DEL	DUP	INV
<0.01 AF 1000G ALL and <0.01 gnomAD	5492	1122	105	137
AnnotSV ranking > 3	274	164	17	18
Total length >= 50 bp	NA	140	17	18
Shared by all samples in a group	154 (HRLS: 66; IRLS: 39; LRLS: 49)	133 (HRLS: 48; IRLS: 46; LRLS: 39)	4 (HRLS: 1; IRLS: 0; LRLS: 3)	13 (HRLS: 4; IRLS: 4; LRLS: 5)
Shared by all samples under a particular group	35 (HRLS: 23; IRLS: 4; LRLS: 8)	18 (HRLS: 7; IRLS: 4; LRLS: 7)	2 (HRLS: 0; IRLS: 0; LRLS: 2)	2 (HRLS: 0; IRLS: 1; LRLS: 1)
Variant’s location according to AnnotSV				
Variants annotation according to hit from RefSeq	BND	DEL	DUP	INV
Intronic	26	17	1	0
Exonic	2	0	0	0
txStart-txEnd	0	0	0	2
NA	7	1	1	0

RefSeq: reference sequence database; NA: not applicable; AF: allele frequency; 1000 G: 1000 genomes project for all individuals in this release; high risk to LS group (HRLS); intermediate risk to LS (IRLS); low risk to LS (LRLS); bp: base pair; BND: breakend; DEL: deletion; DUP: duplication; INV: inversion; gnomAD: Genome Aggregation Database; txStart-txEnd: Transcript Start-Transcript End.

**Table 5 cancers-15-04074-t005:** Classification of structural variants by gene and cancer impact according to pedigree groups.

Structural Variants								
Genes	Detected Variants Stratified by Pedigree Groups				AF	Location	Cancer Related	CRC Related
	AnnotSV ID	LRLS	IRLS	HRLS				
*ADAM10*	Variants Count	2	-	-				
	*15_58912864_58912865_BND_1*	Yes	-	-	﻿NA	INT	-	-
	*15_58913242_58913243_BND_1*	Yes	-	-	NA	INT		
*ATP11A*	Variants Count	1	-	1				
	*13_113518019_113519165_DEL_1*	Yes	-	-	NA	INT	-	-
	*13_113499584_113500055_DEL_1*	-	-	Yes	NA	INT		
*GIGYF2*	Variants Count	2	-	-				
	*2_233668385_233668386_BND_1*	Yes	-	-	NA	INT	-	-
	*2_233668758_233668759_BND_1*	Yes	-	-	NA	INT		
*LINC01137*	Variants Count	2	-	-				
	*1_37938257_37938258_BND_1*	Yes	-	-	NA	INT	-	-
	*1_37938376_37938377_BND_1*	Yes	-	-	NA	INT		
** *RELN* **	Variants Count	-	-	1				
	** *7_103463079_103463080_BND_1* **	**-**	**-**	**Yes**	**0.0001**	**INT**	**Leukemia; Lung; Hepatocellular**	**Gastric**
	** *7_103463462_103463463_BND_1* **	**-**	**-**	**Yes**	**0.0001**	**INT**		
*DIP2C*	Variants Count	1	-	-				
	*10_523437_523544_DEL_1*	Yes	-	-	﻿NA	INT	-	-
*WDR37*	Variants Count	1	-	-				
	*10_1164005_1164234_DEL_1*	Yes	-	-	﻿NA	INT	-	-
*DCAKD*	Variants Count	1	-	-				
	*17_43129091_43129429_DEL_1*	Yes	-	-	NA	INT	-	-
*CDH4*	Variants Count	1	-	-				
	*20_60216779_60217071_DEL_1*	Yes	-	-	0.0003	INT	-	-
*MOV10L1*	Variants Count	1	-	-				
	*22_50585735_50585941_DEL_1*	Yes	-	-	NA	INT	-	-
*DNA2*	Variants Count	1	-	-				
	*10_70222778_70222779_BND_1*	Yes	-	-	NA	INT	-	-
*SBF2*	Variants Count	1	-	-				
	*11_10293916_10293917_BND_1*	Yes	-	-	NA	INT	-	-
*ANO5*	Variants Count	1	-	-				
	*11_22214883_22214884_BND_1*	Yes	-	-	NA	EX	-	-
*CHPT1*	Variants Count	1	-	-				
	*12_102107161_102107162_BND_1*	Yes	-	-	NA	INT	-	-
*MYO5B*	Variants Count	1	-	-				
	*18_47698458_47698459_BND_1*	Yes	-	-	NA	INT	-	-
*PLCB1*	Variants Count	1	-	-				
	*20_8414039_8414040_BND_1*	Yes	-	-	0.0005	INT	-	-
*APOL1*	Variants Count	1	-	-				
	*22_36651344_36651345_BND_1*	Yes	-	-	0.0015	INT	-	-
*NOP14*	Variants Count	1	-	-				
	*4_2941531_2941532_BND_1*	Yes	-	-	NA	INT	-	-
*MRPS18A*	Variants Count	1	-	-				
	*6_43655533_43655534_BND_1*	Yes	-	-	NA	EX	-	-
*CNTNAP2*	Variants Count	1	-	-				
	*7_147571596_147571597_BND_1*	Yes	-	-	NA	INT	-	-
*DPP6*	Variants Count	1	-	-				
	*7_153760690_153760691_BND_1*	Yes	-	-	NA	INT	-	-
*B4GALT1*	Variants Count	1	-	-				
	*9_33130549_33130550_BND_1*	Yes	-	-	NA	INT	-	-
** *IRS2* **	Variants Count	-	1	-				
	** *13_110418067_110419056_DEL_1* **	**-**	**Yes**	**-**	**NA**	**INT**	**Esophageal**	**CRC; Intestinal; Stomach**
*RAP1GAP2*	Variants Count	-	1	-				
	*17_2904534_2904873_DEL_1*	-	Yes	-	NA	INT	**-**	**-**
*ASIC2*	Variants Count	-	1	-				
	*17_31596693_31596759_DEL_1*	-	Yes	-	NA	INT	-	-
*MMP20*	Variants Count	-	1	-				
	*11_102472245_102472246_BND_1*	-	Yes	-	NA	INT	-	-
*KCNIP4*	Variants Count	-	1	-				
	*4_20933624_20933792_DEL_1*	-	Yes	-	NA	INT	-	-
*ADGRE4P*	Variants Count	-	1	-				
	*19_6987869_6987870_BND_1*	-	Yes	-	0.0007	INT	-	-
*DYNLRB1*	Variants Count	-	1	-				
	*20_33116231_33116232_BND_1*	-	Yes	-	NA	INT	-	-
*DTX2*	Variants Count	-	1	-				
	7_76128462_76128463_BND_1	-	Yes	-	NA	INT	-	-
*FIRRE*	Variants Count	-	1	-				
	*X_130813255_130974327_INV_1*	-	Yes	-	NA	TX	-	-
*BCCIP*	Variants Count	-	-	1				
	*10_127513335_127513754_DUP_1*	**-**	-	Yes	NA	INT	-	-
*MGAT5*	Variants Count	-	-	1				
	*2_134966704_134970130_DEL_1*	-	-	Yes	NA	INT	-	-
*COL18A1*	Variants Count	-	-	1				
	*21_46930863_46930934_DEL_1*	-	-	Yes	NA	INT	-	-
*FOXP1*	Variants Count	-	-	1				
	** *3_71242366_71242638_DEL_1* **	**-**	**-**	**Yes**	**NA**	**INT**	**Bladder; Endometrial; Lung; Salivary Gland; Breast; Skin; Hepatobiliary; Prostate; Glioma**	**Esophagogastric; Gastrointestinal; CRC**
*CTNND2*	Variants Count	-	-	1				
	*5_11816774_11817546_DEL_1*	-	-	Yes	NA	INT	-	-
*FSTL4*	Variants Count	-	-	1				
	*5_132918980_132924990_DEL_1*	-	-	Yes	NA	INT	-	-
*DLGAP2*	Variants Count	-	-	1				
	*8_1047119_1047802_DEL_1*	-	-	Yes	0.0001	INT	-	-
*PPP2R5A*	Variants Count	-	-	1				
	*1_212472702_212472703_BND_1*	-	-	Yes	NA	INT	-	-
*BMS1P4*	Variants Count	-	-	1				
	*10_75489277_75489278_BND_1*	-	-	Yes	NA	INT	-	-
*IFITM3*	Variants Count	-	-	1				
	*11_308031_320995_INV_1*	-	-	Yes	NA	TX	-	-
** *RRAS2* **	Variants Count	-	-	1				
	** *11_14348706_14348707_BND_1* **	**-**	**-**	**Yes**	**NA**	**INT**	**Breast; Ovarian**	-
*PRKRA*	Variants Count	-	-	1				
	*2_179314967_179314968_BND_1*	-	-	Yes	NA	INT	-	-
Structural variants with no overlapping genes								
*NA*	Variants Count	6	0	3				
	*17_41438043_41440177_DEL_1*	Yes	-	-	NA	NA	NA	NA
	*11_1961293_1961294_BND_1*	Yes	-	-	NA	NA		
	*17_38679442_38679443_BND_1*	Yes	-	-	NA	NA		
	*19_17459958_17459959_BND_1*	Yes	-	-	NA	NA		
	*6_28863601_28863602_BND_1*	Yes	-	-	0.0002	NA		
	*6_28863898_28863899_BND_1*	Yes	-	-	0.0002	NA		
	*17_41436517_41442619_DUP_1*	**-**	**-**	Yes	NA	NA		
	*9_107816642_107816643_BND_1*	**-**	**-**	Yes	NA	NA		
	*9_107817348_107817349_BND_1*	**-**	**-**	Yes	NA	NA		

AF: 1000 G Phase 3 all population allele frequency; row in bold: gene previously described as associated with cancer; CRC: colorectal cancer; ID: identification; INT: intronic; EX: exonic; NA: not applicable; txStart-txEnd: Transcript Start-Transcript End; high risk to LS group (HRLS); intermediate risk to LS (IRLS); low risk to LS (LRLS).

## Data Availability

The datasets used and analyzed during the current study are available from the corresponding author upon reasonable request.

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
