# Peer review of "Variant Characterization of a Representative Large Pedigree Suggests “Variant Risk Clusters” Convey Varying Predisposition of Risk to Lynch Syndrome"

_cancers, 2023, doi:10.3390/cancers15164074_

Round 1

Reviewer 1 Report

Lynch syndrome is one of the most common hereditary syndromes of cancer predisposition and is associated with an increased risk of colorectal and endometrial cancer, as well as many other cancers.

Therapeutic prospects for Lynch syndrome depend heavily on early diagnosis; however, LS remains underdiagnosed in the population. Accurate diagnosis is inextricably linked to the correct clinical interpretation of individual variants.

The complexity of the classification of variants is due to the fact that variants of unknown meaning are rare in the population and lack phenotypic information about specific variants and that testing of individual variants is challenging, expensive and slow.

This variant characterization study demonstrates that large pedigree investigations provide important evidence supporting the hypothesis that different “variant risk clusters” can convey different mechanisms of risk and oncogenesis of LS-CRC even within the same pedigree.

In the literature there are many reviews on LS and few papers for which this study becomes import to implement knowledge about this syndrome.

Author Response

We appreciate the reviewer’s positive comments.

Reviewer 2 Report

The authors present genomic sequencing on a large Tunisian family with multiple members with colon cancer. They claim that this family has Lynch syndrome with no germline mutation clearly identified. They stratify risk of colon cancer based on degree of relation from those family members with cancer. A more relevant indicator is which family members harbor a MLH1/MSH2/MSH6/PMS2 or EPCAM variant. 

It is unclear if this large family has Lynch syndrome, therefore any subsequent analysis and reference to Lynch syndrome modifiers is unfounded. This paper is likely just genome sequencing of an interesting family with multiple cancers. 

None 

Author Response

Reviewer:   The authors present genomic sequencing on a large Tunisian family with multiple members with colon cancer. They claim that this family has Lynch syndrome with no germline mutation clearly identified. They stratify risk of colon cancer based on degree of relation from those family members with cancer. A more relevant indicator is which family members harbor an MLH1/MSH2/MSH6/PMS2 or EPCAM variant. 

Authors:   We appreciate the reviewer’s comments. The Lynch syndrome within this family was tested and confirmed by pathologists during the routine clinical work for patients with a family predisposition. Most of the patients, including the three patients we recruited and sequenced in this study, underwent Sanger somatic DNA sequencing from biopsies taken during a screening colonoscopy, and positive mutations for MLH1, MSH2, MSH6, PMS2 and EPCAM genes were found. In this study, we selected and identified the family as a Lynch syndrome family according to the clinical data provided by oncologists and pathologists. In addition, we performed whole genome sequencing with only 30X coverage in our study with a small number of samples (11), which may influence the detection of germline mutations that could be more easily detected by whole exome sequencing with higher coverage.

Reviewer: It is unclear if this large family has Lynch syndrome, therefore any subsequent analysis and reference to Lynch syndrome modifiers is unfounded. This paper is likely just genome sequencing of an interesting family with multiple cancers. 

Authors:     Thank you for this constructive comment. A sentence in the Material and Methods section under Sample/data collection was added to the manuscript explaining how the Lynch syndrome was confirmed in our study.

Thank you for your consideration. We hope that the revised manuscript is accepted for publication in Cancers.

Reviewer 3 Report

Rare Hereditary Population study is needed for in the era of precision medicine.

Rare Hereditary Population study is needed for in the era of precision medicine.

Author Response

Dear Reviewer,

We are thankful for you reviewing our manuscript, “Variant characterization of a representative large pedigree suggests ‘variant risk clusters’ convey varying predisposition of risk to Lynch Syndrome” by Barbirou et al., as an original research article for publication in Cancers. We also greatly appreciate your complimentary comments and suggestions. We have carried out the modifications you suggested and revised the manuscript accordingly.

Please find below a point-by-point response to your comments. We hope that you find our responses satisfactory and that the manuscript may now be acceptable for publication. Changes made in the manuscript are highlighted in red. The revision has been developed in consultation with all co-authors, and each author has given approval to the final form of this revision.

Reviewer:   Rare Hereditary Population study is needed for in the era of precision medicine. h this study becomes import to implement knowledge about this syndrome.

Authors:     We appreciate the reviewer’s positive comments.

Thank you for your consideration. We hope that the revised manuscript is accepted for publication in Cancers.

Reviewer 4 Report

the manuscript is based on a very topical and interesting theme: the cooperation of variants in multiple cancer predisposition genes as a cause of the genetic susceptibility to develop tumors present in some families. The results are well described and the tables are complete. However I think the discussions need to be reworked. The rs63750539 variant of the MLH1 gene has been repeatedly described and reported in the literature as a probably pathogenic variant and in this family it is present in all three affected subjects analysed, while it was not found in subjects belonging to the other two groups. Therefore, I think this topic should be addressed in the discussions and only later discuss the effect of the other variants identified in the rest of the genome...

Author Response

Dear Reviewer,

We are thankful for you reviewing our manuscript, “Variant characterization of a representative large pedigree suggests ‘variant risk clusters’ convey varying predisposition of risk to Lynch Syndrome” by Barbirou et al., as an original research article for publication in Cancers. We also greatly appreciate your complimentary comments and suggestions. We have carried out the modifications you suggested and revised the manuscript accordingly.

Please find below a point-by-point response to your comments. We hope that you find our responses satisfactory and that the manuscript may now be acceptable for publication. Changes made in the manuscript are highlighted in red. The revision has been developed in consultation with all co-authors, and each author has given approval to the final form of this revision.

Reviewer:   The manuscript is based on a very topical and interesting theme: the cooperation of variants in multiple cancer predisposition genes as a cause of the genetic susceptibility to develop tumors present in some families.

Authors:     We appreciate the reviewer’s positive comments.

 Reviewer:   The results are well described and the tables are complete.

Authors:     We appreciate the reviewer’s positive comments.

 Reviewer:  However I think the discussions need to be reworked. The rs63750539 variant of the MLH1 gene has been repeatedly described and reported in the literature as a probably pathogenic variant and in this family it is present in all three affected subjects analysed, while it was not found in subjects belonging to the other two groups. Therefore, I think this topic should be addressed in the discussions and only later discuss the effect of the other variants identified in the rest of the genome.

 Authors:    Thank you for this constructive comment. The discussion section of the manuscript has been carefully reviewed and the rs63750539 variant of the MLH1 gene has been discussed according to the literature and your suggestion.

Thank you for your consideration. We hope that the revised manuscript is accepted for publication in Cancers.

Sincerely,

Round 2

Reviewer 2 Report

The revisions the authors present, do not address if this Tunisian family has Lynch syndrome.

The revision: “Lynch syndrome status within this family was tested and confirmed during the routine clinical work by the detection of positive mutations for MLH1, MSH2, MSH6, PMS2 and EPCAM genes through Sanger somatic DNA sequencing from biopsies taken during a screening colonoscopy.” Does not confirm if the family has Lynch syndrome.  Firstly somatic DNA sequencing from a biopsy, is not how Lynch syndrome is diagnosed.  Lynch syndrome is diagnosed from germline testing from a non-somatic tissue such as blood, buccal, saliva or skin biopsy.  Furthermore, a family with Lynch syndrome usually has a single germline variant in one of the five Lynch syndrome genes, and not all five of the genes.

Without knowing the germline variant causing Lynch syndrome, further analysis of the family pedigree with genome sequencing is not meaningful.  The family member who harbor the Lynch variant need to be clearly indicated in the pedigree (Figure 1).  Gene positivity is more predictive of cancer risk, and without known who are gene positive, the high/intermediate/low risk classification is not useful.

could be improved

Author Response

Authors: In response to the major revision requested, we find it necessary to revisit the medical records of the human subjects that were involved in our study. These records are associated with a family that we recruited several years prior to the implementation of our current numeric informatic system for maintaining medical records. As such, the data we require resides in a physical archive. This situation requires the assistance of a study participant (a medical oncologist) who would need to physically locate and extract and review the necessary data from the archive.

                   After reviewing the medical records of the study family and several team meetings between the authors and colleagues specialized in Lynch Syndrome, we found that the family has been identified with Lynch syndrome according to two criteria that are used in Tunisia's routine clinical work:

  1. The clinical data of the CRC patients met both the Amsterdam criteria II established by the International Collaborative Group on HNPCC and the original Bethesda guidelines.
  2. Germline genetic testing of the blood based on PCR amplification and direct sequencing of the MMR genes (MLH1, MSH2 and MSH6) coding regions showed a single deleterious germline alteration affecting the MLH1 gene (mutation: c.-168_c.116+713del). This same mutation was detected in the three subjects from the HRLS group that underwent germline testing; concerning the other subjects included in the study’s IRLS and LRLS groups, there is no germline data to provide.

We included this information in the Material and Methods section under Sample/data collection to the manuscript explaining how the Lynch syndrome was confirmed in our study as well as Figure 1 as requested.

Thank you for your consideration. We hope that the revised manuscript is accepted for publication in Cancers.

Sincerely,

Corresponding Author; Mouadh Barbirou, MS.c, PhD.

Thomas Jefferson University, 1025 Walnut St, Suite 727, Philadelphia, PA 19107, Tel: 215-503-6120.

E-mail: Mouadh.Barbirou@jefferson.edu ; mouadh_barbirou@yahoo.com

Reviewer 4 Report

Discussions should be about the significance of the variant identified in the MLH1 gene, rather than the association of the gene with the disease that has been known for so long. For example, discuss whether the identified missense variant in the MLH1 gene could be the cause of Lynch syndrome in this family and, then discuss the possible existence of other causative genes of Lynch syndrome. Finally, also discuss the significance of the variants of uncertain pathogenetic significance in the MMR genes as responsible for Lynch syndrome and then the possible synergistic effect with other genes on the onset of the tumor typical of Lynch syndrome.

Author Response

Authors:     We appreciate the reviewer’s comments. In this study, we are focusing on the combination of multiple variants which may lead to different levels of cancer risk. Instead of sequentially discussing the variants one-by-one, as typically done in traditional germline studies, we intend to show that the cancer risk may be affected by a combination of known and novel variants, each playing distinct roles across various cancerous and non-cancerous pathways. This novel perspective offers an avenue for researchers to explore the potential significance of emerging variant clusters.

Thank you for your consideration. We hope that the revised manuscript is accepted for publication in Cancers.

Sincerely,

Corresponding Author; Mouadh Barbirou, MS.c, PhD.

Thomas Jefferson University, 1025 Walnut St, Suite 727, Philadelphia, PA 19107, Tel: 215-503-6120.

E-mail: Mouadh.Barbirou@jefferson.edu ; mouadh_barbirou@yahoo.com.

Round 3

Reviewer 2 Report

The authors now have confirmed that this family has a germline MLH1 variant.  The authors should provide the ACMG interpretation of pathogenicity of the variant.  However, without having all members of the family tested for this variant the conclusions are not meaningful.  The stratification of cancer risk must be done in the context of MLH1 carriers and non-carriers.  If family members are not available for testing, some inferences on which family members are carriers can be done with simple medical genetics (e.g. obligate carriers).  Furthermore, whole genome sequencing was done on 11 family members, therefore the MLH1 variant should be detected.

At this time, this study is simply examining an MLH1 family with some members undergoing WGS.  The conclusions about genetic modifiers in Lynch are not meaningful unless all family members have MLH1 characterization completed.  Otherwise, this study is population-based WGS and the conclusions are not scientifically sound.

no comments

Author Response

Reviewer:   The authors now have confirmed that this family has a germline MLH1 variant.  The authors should provide the ACMG interpretation of pathogenicity of the variant.  However, without having all members of the family tested for this variant the conclusions are not meaningful.  The stratification of cancer risk must be done in the context of MLH1 carriers and non-carriers.  If family members are not available for testing, some inferences on which family members are carriers can be done with simple medical genetics (e.g. obligate carriers).  Furthermore, whole genome sequencing was done on 11 family members, therefore the MLH1 variant should be detected.

Authors:    In response to your previous concern, we consulted with all authors and colleagues about how this family was classified as LS, and we have provided the clinical (the family met Amsterdam criteria II) and molecular (germline testing showed a single deleterious deletion in MLH1 gene, mutation: c.-168_c.116+713del) evidence. This pathogenicity interpretation is used worldwide by clinicians to confirm LS. However, at this stage, we cannot test all members of the family. First, because most of the members passed away. Second, there is no routine protocol in Tunisia that can request a specimen collection from a family member without cancer. Subjects included in our study voluntarily agreed and signed a consent form to be tested and included in this study. Despite our efforts to contact most of the family members, many of them they refused to be tested.

Reviewer:   At this time, this study is simply examining an MLH1 family with some members undergoing WGS.  The conclusions about genetic modifiers in Lynch are not meaningful unless all family members have MLH1 characterization completed.  Otherwise, this study is population-based WGS and the conclusions are not scientifically sound.

Authors:    This study is the first to involve WGS in a clinically classified Tunisian family with LS, based on the Amsterdam and MMR genes criteria used in Tunisia. By offering a novel viewpoint and encouraging researchers across North Africa and nearby countries to consider the LS molecular classification used in the region, it will promote further studies on family-based cancers, improving the understanding of the germline predisposition of this specific population. Based on this and other socioeconomic criteria, we believe that our conclusion “According to the familial definition of LS risk in the pedigree members, we identified three ‘variant risk clusters’ associated with the High-, Intermediate-, and low LS CRC risk groups in the pedigree” supports that the application of HTS technology will efficiently provide further insights into the etiology of hereditary cancer and a substantial opportunity to improve diagnosis, prognosis, and predict metastasis.

However, we took your comment into consideration and included a paragraph at the end of the discussion explaining the weaknesses of not testing all of the family members.

Reviewer 4 Report

authors should check the veracity of some sentences such as the following, "The literature suggests that LS is caused by genetic and epigenetic variants sporadically found in genes, such as MMR and APC associated with flat intra-mucosal neoplastic lesions [24][25] ". .....LS is never associated with variants in the APC gene..... moreover, when the author refers to the family of pedigree under examination, he must not speak of a diagnosis of LS but of a clinical suspicion of LS .. .. The diagnosis is made only in the presence of a definitely pathogenetic variant... Also this sentence is superfluous, "A different study identified a heterozygous novel MLH1 mutation (c.206delG) in a 40-year-old male with early- onset CRC [29 ],” truncating variants in MLH1 have long been known to cause LS... The discussion section needs to be carefully double-checked...

Author Response

Reviewer:   authors should check the veracity of some sentences such as the following, "The literature suggests that LS is caused by genetic and epigenetic variants sporadically found in genes, such as MMR and APC associated with flat intra-mucosal neoplastic lesions [24][25] ". .....LS is never associated with variants in the APC gene.....

Authors:     We appreciate the reviewer's comments, which helped us correct and improve our manuscript. We corrected the typo related to the MMR genes (MLH1, MSH2, MSH6, PMS2, and EPCAM), and after checking references 24 and 25, we made the necessary corrections in the manuscript.

Reviewer: moreover, when the author refers to the family of pedigree under examination, he must not speak of a diagnosis of LS but of a clinical suspicion of LS .. .. The diagnosis is made only in the presence of a definitely pathogenetic variant... Also this sentence is superfluous, "A different study identified a heterozygous novel MLH1 mutation (c.206delG) in a 40-year-old male with early- onset CRC [29 ],” truncating variants in MLH1 have long been known to cause LS... The discussion section needs to be carefully double-checked...

Authors:     We agree with the reviewer and have double-checked the discussion. We removed this sentence as it does not contribute any potential information and switched the use of “diagnosis” to “clinical suspicion” in our discussion.
